# Performances of Limited Area Models for the WORKLIMATE Heat–Health Warning System to Protect Worker’s Health and Productivity in Italy

**DOI:** 10.3390/ijerph18189940

**Published:** 2021-09-21

**Authors:** Daniele Grifoni, Alessandro Messeri, Alfonso Crisci, Michela Bonafede, Francesco Pasi, Bernardo Gozzini, Simone Orlandini, Alessandro Marinaccio, Riccardo Mari, Marco Morabito

**Affiliations:** 1Institute of Bioeconomy—National Research Council (IBE-CNR), 50019 Sesto Fiorentino, Italy; daniele.grifoni@ibe.cnr.it (D.G.); alfonso.crisci@ibe.cnr.it (A.C.); francesco.pasi@ibe.cnr.it (F.P.); gozzini@lamma.toscana.it (B.G.); riccardo.mari@ibe.cnr.it (R.M.); marco.morabito@ibe.cnr.it (M.M.); 2Tuscany Region, LaMMA Consortium, 50019 Sesto Fiorentino, Italy; 3Centre of Bioclimatology—University of Florence (UNIFI), 50100 Florence, Italy; simone.orlandini@unifi.it; 4Occupational and Environmental Medicine, Epidemiology and Hygiene Department, Italian Workers’ Compensation Authority (INAIL), 00078 Rome, Italy; m.bonafede@inail.it (M.B.); a.marinaccio@inail.it (A.M.); 5Department of Agriculture, Food, Environment and Forestry (DAGRI), University of Florence, Piazzale delle Cascine 18, 50144 Florence, Italy

**Keywords:** occupational health and safety, wet-bulb globe temperature (WBGT), climate change, high-resolution forecasts, personalized forecasts for workers, limited area model (LAM), meteorological model performance

## Abstract

Outdoor workers are particularly exposed to climate conditions, and in particular, the increase of environmental temperature directly affects their health and productivity. For these reasons, in recent years, heat-health warning systems have been developed for workers generally using heat stress indicators obtained by the combination of meteorological parameters to describe the thermal stress induced by the outdoor environment on the human body. There are several studies on the verification of the parameters predicted by meteorological models, but very few relating to the validation of heat stress indicators. This study aims to verify the performance of two limited area models, with different spatial resolution, potentially applicable in the occupational heat health warning system developed within the WORKLIMATE project for the Italian territory. A comparison between the Wet Bulb Globe Temperature predicted by the models and that obtained by data from 28 weather stations was carried out over about three summer seasons in different daily time slots, using the most common skill of performance. The two meteorological models were overall comparable for much of the Italian explored territory, while major limits have emerged in areas with complex topography. This study demonstrated the applicability of limited area models in occupational heat health warning systems.

## 1. Introduction

Climate change projections indicate that most people who inhabit our planet will experience more recurrent natural hazards [1], and particularly, intense and longer-lasting heatwave periods over the coming decades [2]. The world of work, especially that carried out outdoors, is intimately connected with the natural environment and climate conditions. The increase of environmental temperature directly affects the occupational sector in a generally negative way [3,4]. Looking ahead, the heat stress phenomenon, that the National Institute for Occupational Safety and Health (OSHA) defines as the sum of the heat generated in the body (metabolic heat) plus the heat gained from the environment minus the heat lost from the body to the environment [5], will become an even more important issue, impacting on the health of workers and reducing the total number of working hours. In particular, during heatwaves, outdoor workers are those who present the greatest sun exposition, dehydration, and heat stress that can lead directly to heat-related illnesses [6,7] as well as an increased risk of accidents happening because of the tiredness and lack of concentration due to working in the heat [8,9]. These effects are expected to increase over the next few years not only because of climate change, but also because of demographic changes in the working population. The increasing average age of the working population affects various components of the physical work capacity, including aerobic power and capacity, muscular strength, and tolerance of thermal stress [10]. In addition, the increasing number of immigrant workers represents an additional critical factor due to cultural aspects (religious, linguistic, adaptation to local conditions). Immigrants reveal a different perception of the heat risk and consequently a greater vulnerability [11,12]. It is also important to note that workers involved in outdoor activities, especially in agriculture and construction sectors, often wear personal protective clothing and equipment that significantly increases the heat stress by limiting the body heat loss. The heat stress vulnerability of a worker is strictly individual and therefore depends on a multiplicity interconnected factors: work environment, work effort, physical characteristics, state of health, hydration status, age, and type of clothing worn. In light of this situation, it is fundamental to increase adaptation strategies with the aim to mitigate the effects heat conditions at different temporal scales (few days to decades), also including local microclimatic monitoring and developing warning systems that are also representing the priorities of both World Meteorological Organization (WMO) and World Health Organization (WHO). For these reasons, in recent years, a great number of heat-health warning systems (HHWSs) have been developed for the general population [13], and in particular for vulnerable groups including workers [14,15].

HHWSs generally do not use single standard parameters, e.g., air temperature and humidity, wind speed, or solar radiation, but a combination of them expressed as an index, to describe in detail the thermal stress induced by the outdoor environment on the human body. For these reasons, a great number of biometeorological indices have been developed and find application in various fields. The empirical Wet Bulb Globe Temperature (WBGT) according to UNI EN ISO 7243 [16] and the rational Predicted Heat Strain (PHS) according to UNI EN ISO 7933 [17] are currently the only systems developed at an international level for an objective assessment of heat stress referring to groups of workers. In particular, the WBGT, being an empirical index and easier to apply, is used precisely for a first screening of heat stress on workers and is therefore suitable for applications in the forecasting meteorological field. The Wet-Bulb Globe Temperature (WBGT) index [18,19] represents the international reference among heat stress indices for work activity assessments [5,15,16,20] because it responds to the needs of the occupational sector that are different compared to the general population and other vulnerable groups. Recently, the WBGT, was also chosen and used as the heat stress indicator in the “HEAT-SHIELD occupational warning system” platform [15] within the frame of the European HEAT-SHIELD Project (HORIZON 2020, research and innovation program under the grant agreement 668786). In particular, it was the first operational website platform providing personalized short- and long-term heat warning (up to 46 days) with also hydration and work/break schedule recommendations (up to five days) to safeguard workers’ health and productivity. The HEAT-SHIELD HHWS is currently the only warning system addressed to workers that provides forecasts up to medium–long term range, using a probabilistic meteorological model calibrated with observations on specific locations. However, this system has some limitations: provides information with a low temporal resolution (daily forecast without any sub-daily detail) because it is based on a monthly ensemble forecasts model (ECMWF) without detailed intra-daily forecast. It is location-specific because the forecasts are available only for 1800 European locations where downscaling and bias-correction procedures are applied using observed data. It is available as a web service and not as APP.

These HHWSs systems are based on the outputs of different weather forecast systems from high-resolution models to probabilistic ensembles or to a combination of them [15]. In the case of Europe, the national agencies run their own simulations or use those from the European Center for Medium-Range Weather Forecasts (ECMWF) or a combination of both. There are also ready-to-use products from the ECMWF such as the EFI (extreme forecast index) of temperature, indicating how extreme predicted temperatures are [13]. There are several studies on the verification of the parameters predicted by meteorological models [21,22,23,24,25,26], but very few relating to the validation of heat stress indicators [15,27].

To try to fill the gaps in the HEAT-SHIELD HHWS, an Italian Project (WORKLIMATE) approved under the BRIC-INAL 2019 funding is under development. It is focused on the estimation of social costs of accidents at work and on the development of heat-related adaptation strategies for workers also accounting for qualitative approaches. One of the main objectives of the project will be to develop a first experimental version of an occupational HHWS for Italy, taking into account also epidemiological aspects. The HHWS will be represented by a first high resolution experimental version of Web Forecasting Platform (https://www.worklimate.it/en/maps-choice/shade-intense-physical-activity/; accessed on 18 September 2021) and a mobile web app with personalized heat-stress-risk based on the worker’s characteristics and on the work environment (i.e., workers exposed to the sun or in shaded areas). In particular, WORKLIMATE will try to respond to several occupational needs providing a specific and detailed personalized HHWS useful for worker and various stakeholders with detailed intra-daily information (per time slots) in the short term (forecast up to five days) concerning the heat risk level and behavioral suggestions (hydration and breaks recommended) to reduce the impact of the heat on different occupational sectors. Furthermore, the recommendations provided will also take into account the presence of some individual vulnerability/susceptibility factors.

To be able to meet these requirements, the goal is to use, in the WORKLIMATE operational chain, a limited area meteorological model to achieve a high scale of analysis (less than 10 km) and temporality (sub-daily detail of the forecasts).

In particular, the performances of two limited area models operational at the “Environmental Modelling and Monitoring Laboratory for Sustainable Development (LaMMA Consortium)” were tested on several locations along Italy over a period of about three summer seasons and their possible use in the operational chain will be discussed highlighting strengths and weaknesses.

## 2. Materials and Methods

### 2.1. Methodology

The comparison between model outputs and weather stations data was carried out over the period May–September and for 4 time slots: 0–6, 6–12, 12–18, 18–24 (daylight saving time). Each time slot was analyzed both considering all its hourly data and its maximum value. In the paper only the results of hottest period of the day, time slot 12–18, were showed. The analyses were performed for each weather station and for both Day2 (tomorrow forecast) and Day3 (after tomorrow forecast). Day1 (today) data are not shown because it is not fully suitable for an alert system that must provide information at least 24 h in advance. Results are presented both per station and as an average value for homogeneous geographical areas.

### 2.2. Meteorological Observation Dataset

Hourly meteorological data (air temperature, air humidity, and wind speed) of about 40 Italian weather stations were collected and archived in order to verify the performances of different meteorological models. The meteorological stations have been chosen in order to represent most of the climatological characteristics of the most populated Italian areas compatibly with data availability.

The main sources of data were the Regional Hydrologic Services of Tuscany and Umbria (SIR) and National Weather Service (AM). These services are responsible for the maintenance and data validation and each weather station was installed in accordance with the rules of the World Meteorological Organization [28,29]. Concerning solar radiation, the METEOSAT satellite estimation from LSA SAF products belonging to EUMETSAT (https://www.eumetsat.int/lsa-saf; accessed on 21 September 2021) was used due to the difficulties in obtaining reliable ground data. Meteorological hourly data were collected for the period 1 July 2018–7 August 2020.

After a first check on the collected dataset, only 28 stations showed continuity and good quality of data during the period 2018–2020 and in particular during the hottest months of the year (from May to September) and in the daytime slots (06:00–12:00 and 12:00–18:00 in daylight saving time). The distribution of the stations is not homogeneous, however sufficient to highlight possible critical issues. The distribution of the weather stations over Italy is shown in Figure 1.

The 28 weather stations are also listed in Table 1 where they have been classified by three geographical macro-areas: North inland plain areas (A), Coastal areas (B); Central-south inland areas (C). For each location, latitude, longitude, and altitude are shown.

Concerning the macroarea A, Bolzano was not included during the calculations of the average skill scores by area, because contrary to the others locations it was in a very narrow valley surrounded by very high mountains (very complex topography), and for this reason the model reconstructs a very higher elevation (about 1050 a.s.l) than the real one (262 m a.s.l). However, it was used to compare its skill scores with those of the other location of the area.

### 2.3. Meteorological Forecast Model Dataset

Between the limited area models available at the “Environmental Modelling and Monitoring Laboratory for Sustainable Development- LaMMA Consortium”, Bolam and Moloch models were chosen for the comparison. LaMMa (https://www.lamma.rete.toscana.it; accessed on 21 September 2021) is a public consortium between the Tuscany Region and the National Research Council which carries out activities related to observation systems and meteorological modeling at different spatial scales. Furthermore, the LaMMA provides meteorological forecasts to the Civil Protection and carries out research activities in various fields, including the climatological one. The Bolam model [30,31] is a hydrostatic meteorological model, continuously developed at CNR-ISAC (Bologna, Italy) in 1992. The main prognostic variables are the wind components, the absolute temperature, the surface pressure, the specific humidity, and the turbulent kinetic energy. The surface layer and the planetary boundary layer are modelled according to the similarity theory [32], with a mixing-length based turbulence closure model, to parameterize the turbulent vertical diffusion of momentum, heat and moisture. The turbulence closure is of order 1.5 [33], in which the turbulent kinetic energy is predicted. The Soil Model uses 4–6 layers and computes surface energy, momentum, water and snow balances, heat and water vertical transfer, and vegetation effects at the surface (evapo-transpiration, interception of precipitation, wilting effects etc.) and in the soil (extraction of water by roots). It takes into account the observed geographical distribution of different soil types and soil physical parameter. The atmospheric radiation is computed with a combined application of the global radiation [34] scheme and the ECMWF scheme [35,36]. The model was tested and favorably compared with many other limited area models, in the course of the Comparison of Mesoscale Prediction and Research Experiments [37,38] as well as the MAP (Mesoscale Alpine Programme) field phase [39]. Moloch, on the other hand, is a non-hydrostatic, fully compressible, convection resolving model recently developed at CNR-ISAC in 2000 [40]. The model was employed, among other studies, in the international forecasting demonstration project called MAP-DPHASE, in which many mesoscale high-resolution NWP models were compared in real time (during autumn 2007), especially in relation to QPF (Quantitative Precipitation Forecasting—[41]) and in the European project RISKMED [42]. The two models have surface schemes (Land Surface Model, Planetary Boundary Layer and Radiation) very similar, with the exception of specific differences introduced in Moloch to treat the complex processes characterizing convective systems, and hence behave in similar way in forecasting surface variables (e.g., 2 m temperature and dew point, 10 m winds and windgust, short and long wave radiation). Other differences between the two models present in the different horizontal resolution (7 km for Bolam vs. 2.5 km for Moloch), and in initial and boundary conditions. Davolio et al. [43] reported that Bolam and Moloch have been used for numerous scientific studies and applications, e.g., sensitivity and impact studies, and diagnostics of meteorological phenomena, including severe weather and storms. In addition, they also reported that these models were used in several operational applications.

The operational chain of Bolam is based on initial and boundary conditions provided by the Global Forecast Model (GFS) of the NCEP at 0.25 deg resolution (about 25 km), 2 runs a day (00 and 12 UTC) performed with a lead time of +120 h for 00 run and +132 h for 12 run. The operational chain for Moloch is based on initial and boundary conditions provided other than by GFS (as Bolam) also by the IFS Global Model of the ECMWF at 0.10 deg resolution (about 10 km), 4 runs a day (00, 06, 12 and 18 UTC) performed with a lead time of +84 h for 00 run, +42 h for 06 run, +84 h for 12 run and +54 h for 18 UTC run. In the present study, only the 00 UTC run and the first 72 h of prediction were considered. In this paper, the Bolam model will be called BOL, while Moloch will be called MOL-G and MOL-E, respectively, depending on whether it is initialized with the GFS or with IFS Global Model of the ECMWF.

Hourly model outputs were available from 1 July 2018 to 7 August 2020. Figure 2 shows, for each location, the elevation of the meteorological stations and that of the closest meteorological model grid points (BOL and MOL). The model grid point extraction was performed using the only criterion of the minimum distance from the location without any type of correction.

Figure 3 shows the areas of Italian peninsula where the models grid points have an elevation higher than at least 200 m with respect to that of a digital terrain model (DTM) with a spatial resolution of 90 m (overestimation of the elevation).

### 2.4. Heat Stress Indicator

The Wet Bulb Globe Temperature (WBGT) index was selected as the heat strain indicator for the WORKLIMATE high-resolution heat–health warning system. WBGT was developed in the 1950s as a basis for environmental heat stress monitoring to control heat casualties at military training camps in the USA [5,18] and in particular in a study on heat-related injuries during military training [44]. Today, it represents the most commonly used heat stress index for a first screening of heat stress conditions in workplaces, with recommended rest/work cycles at different metabolic rates clearly specified in the international standard to ensure that the average core body temperature of a worker does not exceed 38°C [16]. The WBGT represents a good compromise between the data forecasted by the meteorological model and the quality/usefulness of the forecast information of the heat risk taking into account the various exposure scenarios to which workers are exposed. WBGT is considered to fulfill the purpose for individualized heat warnings, with customized limits for different workers potentially useful for managing policies against the heat effects. For this reason, it was also chosen and used as the heat stress indicator in the “HEAT-SHIELD occupational warning system” [16] realized within the frame of the European HEAT-SHIELD Project (grant agreement 668786). WBGT is a combination of the following meteorological parameters:-Dry-bulb temperature (Ta), measured with a thermometer shaded from direct heat radiation.-Natural wet-bulb temperature (Tnwb), measured with a wetted thermometer exposed to the actual wind and heat radiation.-Black Globe Temperature (Tg), measured inside a 150mm diameter black globe.

This indicator therefore allows to estimate the thermal stress conditions both of a subject exposed to the direct short-wave radiation (WBGT-sun) and of a subject not directly exposed to direct short-wave radiation (WBGT-shade). For WBGT workplace calculation starting from meteorological data, Lemke and Kjellstrom’s [45] procedure was used and the approach of Bernard and Pourmoghani [46] was also applied [47] for computing WBGT in the shade and in the sun, respectively. These implementations allow the calculation of both the natural wet bulb temperature, that is the largest component (70%) of WBGT, and the black globe temperature (it contributes 20–30% of WBGT) as required by the WBGT formulas starting from air temperature, humidity, wind speed, and solar radiation provided by the weather forecast model [16]. WBGT-sun and WBGT-shade hourly values were also calculated using the limited area models’ meteorological data provided by the LaMMA Consortium. Using the procedure already used in the Heat-Shield forecast system [15], the predicted WBGT value was corrected in WBGTeff to take into consideration the clothing information and then compared with the customized risk threshold (WBGT RAL and WBGT REL for acclimatized and unacclimatized worker, respectively), obtaining the risk level (RL) [16]:WBGT*eff* = WBGT predicted + Clothing Adjustment Value (CAV) as described in ISO 7243(1)
WBGT RAL(°C) = 59.9 − 14.1 log10 MR(2)
WBGT REL(°C) = 56.7 − 11.5 log10 MR(3)
RL (%) = (WBGT*eff*/WBGT RAL (o WBGT REL)) × 100(4)
RL0 (green) = RL(%) ≤ 80(5)
RL1 (yellow) = 80 < RL(%) < 100(6)
RL2 (orange) = 100 < RL(%) < 120(7)
RL3 (red) = RL(%) ≥ 120(8)

The 5-day threshold with critical heat stress conditions (in our case with at least a moderate risk level) was used to define when a worker can be considered acclimatized to heat within a warm season.

The RL (0 not significant; 1 low risk, 2 moderate risk, 3 high risk) were calculated for a standard worker (weight 75 kg, height 175 cm), acclimatized to heat, engaged in intense physical activity, and wearing normal working overalls. The RL predicted (RLP) by the limited area model were then compared with the RL obtained using meteorological parameters (RLO) recorded by weather stations (following in the paper observed data). Obviously, the RL skill scores obtained considering this typology of worker are to be considered purely indicative as they may vary with the characteristics of the worker.

### 2.5. Data Analysis and Forecast Evaluation Metrics

The hourly RLPs and RLOs (both for WBGT-sun and WBGT-shade) were compared using contingency tables (Table 2). For each of the 28 location, contingency tables were created, taking into account the day of forecast and the daily time slot. Each table was then populated with the hourly RLP and RLO pair values. In this way, the table diagonal represents the number of hours with correct forecast, i.e., the hours in which the RLP exactly matches the RLO.

Then, the following skill scores were calculated [48]:-Hit rate (HR): Correct predictions probability (%) on the total of events (including class 0).
$$\mathrm{HR}=\frac{C00+C11+C22+C33}{C00+C10+C20+C30+C01+C11+C21+C31+C03+C13+C23+C33}\;\times \;100$$

-Critical success index (CSI): Correct predictions probability (%) considering only RL ≥ 1.


$$\mathrm{CSI}=\frac{C11+C22+C33}{C10+C20+C30+C01+C11+C21+C31+C02+C12+C22+C32+C02+C13+C23+C33}\;\times \;100$$


-Probability of detection (POD): Correct predictions probability (%) of any class. This skill was calculated for RL1 (POD1), RL2 (POD2), and RL3 (POD3). POD was also calculated, also considering the forecast of a higher class than the observed as correct. This was carried out for both RL1 (POD1x) and RL2 (POD2x).


$$\mathrm{POD}1=\frac{C11}{C10+C11+C12+C13}\;\times \;100$$



$$\mathrm{POD}2=\frac{C22}{C20+C21+C22+C23}\;\times \;100$$



$$\mathrm{POD}3=\frac{C33}{C30+C31+C32+C33}\;\times \;100$$



$$\mathrm{POD}1x=\frac{C11+C12}{C10+C11+C12+C13}\;\times \;100$$



$$\mathrm{POD}2x=\frac{C22+C23}{C20+C21+C22+C23}\;\times \;100$$


-Lack alarm ratio (NA): The probability (%) that if RL0 was predicted, a higher class has been observed instead.


$$\mathrm{NA}=\frac{C10+C20+C30}{C00+C10+C20+C30}\;\times \;100$$


-False alarm ratio (FA): The probability (%) that if RL0 is observed, a higher class has been predicted instead.


$$\mathrm{FA}=\frac{C01+C02+C03}{C00+C01+C02+C03}\;\times \;100$$


-Normalized lack alarm ratio (NA*): Lack alarm probability (%) normalized on the total number of hours analyzed.


$$\mathrm{NA}*\;=\frac{C10+C20+C30}{C00+C10+C20+C30+C01+C11+C21+C31+C02+C12+C22+C32+C03+C13+C23+C33}\times \;100$$


-Normalized false alarm ratio (FA*): False alarm probability (%) normalized on the total number of hours analyzed.


$$\mathrm{FA}*\;=\frac{C01+C02+C03}{C00+C10+C20+C30+C01+C11+C21+C31+C02+C12+C22+C32+C03+C13+C23+C33}\;\times \;100$$


In addition to the skill scores on WBGT expressed in terms of categorical RLs, mean error (ME), mean absolute error (MAE), and root mean square error (RMSE) have been calculated for continuous variables (included WBGT expressed as temperature). The ME is the average of the deviations between predicted (Y) and observed (O) values:$$\mathrm{ME}=\frac{1}{M}\overset{M}{\underset{m=1}{\sum }}\left(\mathrm{Ym}-\mathrm{Om}\right)$$

When ME is 0, it means that the positive and negative deviations between the predicted and observed values balance out. For this reason, a ME equal to zero can be the result of either deviation close to 0, but also the result of positive and negative deviations that balance each other. To evaluate the gap average size in absolute value, the MAE was used:$$\mathrm{MAE}=\frac{1}{M}\overset{M}{\underset{m=1}{\sum }}|\;\left(\mathrm{Ym}-\mathrm{Om}\right)\;|$$

ME equal to 0 associated with an MAE close to zero is the desirable situation. Finally, the RMSE was also calculated, which attributes a greater weight to the largest gaps:$$\mathrm{RMSE}=\sqrt{\frac{1}{M}\overset{M}{\underset{m=1}{\sum }}\left(\mathrm{Ym}-\mathrm{Om}\right)}$$

## 3. Results

### Day2-WBGT Forecast Validation (Period May-September, Time Slot 12–18 All the Hour)

BOL, MOL-E, and MOL-G showed a similar probability of detection of the RL1 (POD1) that represents the most frequent risk level observed and predicted (RLO1 and RLP1) (values close to 50%) during the hottest time slot of the warm period (Table 3).

The average POD1 showed values close to 80% and with the highest values in “coastal areas” (BOL = 84.2% and MOL-G = 84%). If the forecast in a higher class than the observed one is also considered correct (POD1x), the score rises above 90% for almost all models and for all macro-geographical areas. The highest values in this case were in “northern inland plain areas” (MOL-E = 98.1%, MOL-G = 97.2%, and BOL = 96%).

The variability of the skill scores between the different locations of each macro-geographical areas was minimal (data not shown), and for Bolzano despite in an area with a complex orography the skill scores were relatively high and not too different from that of the other north inland locations where POD1 was between 57 and 80% and POD1X between 85 and 90%. Considering the average probability of detection of the RL2 (POD2) the highest values, around 90%, were observed in “northern inland plain areas”, while values in a range of 68–75% and 62–80% were on “coastal areas” and “other central-south inland areas”, respectively. The lowest value of 62% was for BOL. No significant increases were observed in POD2x versus POD2, because almost never a RL3 was predicted considering the worker characteristics previously described. In almost all cases in which RL2 has been observed, there was at least one risk class (RL1 or RL2), and rare exceptions temporarily occurred in areas with particularly complex topography (such as for example in Alpine and Apennine valleys or some coastal areas). Figure 4 shows the probability of detection (POD2) of the Day2-WBGT-shade for the 12:00–18:00 time slot for each location of the three macro-geographical areas.

In “northern inland plain areas” the POD2 values in several locations were around 90% for all models with the exception of Bolzano where BOL was not able to predict RL2, whereas MOL-G and MOL-E in a percentage around 50–60%. In the “coastal areas”, POD2 was generally higher than 60% for several locations and for all models except for Capo Bellavista with POD2 ranging between 63% (MOL-E) and 44% (BOL). In the “other central-southern inland plain areas”, BOL showed rather low POD2 values (<30%) in Foligno and Lamezia, while for MOL-E and MOL-G it was close to 60%. For the other locations in the area, there were no significant differences between the models with POD2 above 60%. Concerning the lack alarm (NA), the lowest number was observed in “northern inland plain areas” (NA = 5.3–8.5%) while a progressive increase was observed moving from “coastal areas” (NA = 11–12%) to “other internal central-southern areas” (NA = 15–24.3%). The highest values were reached for BOL especially in the “other central-southern internal areas (NA = 24.3) (Table 3). A similar pattern was shown by the normalized lack alarms (NA*), but with significantly lower values. Figure 5 shows the normalized lack alarms (NA*) for the forecast of the Day2-WBGT-shade for the 12–18 time slot for each location of the three macro-geographical areas. The highest levels of NA* were found for BOL in areas with greater topographic complexity, well represented by Bolzano and secondarily by Foligno and Lamezia (NA* 38%, 14% and 9% respectively), while elsewhere the scores for different models were similar.

Concerning false alarms (FA and FA*) were greater in the “northern inland plain areas” for MOL-E (FA = 28.2 and FA* = 7.2) and MOL-G (FA = 26.1 and FA* = 7). It is interesting to observe how RL3 for WBGT-shade was almost never predicted or observed in all areas during the analyzed period, making it impossible to calculate the corresponding average POD3 (some had no data). At least for the time slot 12–18, the results obtained considering all its hourly values were not different to those obtained with its maximum value (data not shown).

Mean error (ME), mean absolute error (MAE), and root mean square error (RMSE) calculated on the numerical value of WBGT-shade confirmed a very similar performance of models in predicting WBGT-shade for all macro-geographical areas (Table 4). ME values were positive in “northern inland plain areas” for BOL (0.4), MOL-E (0.7), and MOL-G (0.7). On the contrary, they were slightly negative in the coastal areas and in the “other central-southern inland areas” (−0.8 < ME < 0), highlighting a slight underestimation of the WBGT-shade values in these areas. Moreover, considering these skill scores, Bolzano showed the highest error especially for BOL (ME = −3.5 for BOL and −0.6 for MOL-E). The average mean absolute error for different models and areas was between 1 and 1.4 °C. The skill scores were very similar also considering the maximum time slot value with an underestimation of about 1 °C for BOL in other inland areas.

The results relative to the prediction of WBGT-sun (Table 5), and therefore of the RL for a worker who carries out his activities directly exposed to solar radiation, are very similar to those observed for the WBGT-shade, even with an improvement in performance of the models for the forecast of the RL2.

In particular, POD2 and POD2x showed the highest values in “northern inland plain areas” with percentages close to 90% in all models and the highest value with MOL-E (94.1%). The lowest probability of detection for RL2 was for BOL in “other central-southern inland areas” (78.4%). POD1 and POD1x instead have slightly lower values than the WBGT-shade for all models. Moreover, for the WBGT-sun, although the observed RL3 increased, it was not possible to calculate the average POD3 (some locations had not data). The lack of alarm (NA and NA*) was similar to that observed for WBGT-shade, while the false alarms (FA) were greater (with the highest values in the “northern inland plain areas). However, considering the normalized value (FA*) the differences were significantly reduced. Moreover, for the WBGT-sun, mean error (ME), mean absolute error (MAE), and root mean square error (RMSE) confirmed a very similar performance of the models for all macro-geographical areas (Table 6).

The models showed the best average performances in the “coastal areas” (ME~0) while the highest values of ME (0.9) were for MOL-E in the “northern inland plain areas”. Compared to the WBGT-shade, there were generally no model underestimations in any geographic area.

The forecast for the third day showed values substantially comparable to those that emerged in the evaluation of the performance of the models for the second day (Appendix A).

In general, the WBGT forecast proved more skillful than that of the single meteorological parameters used for its calculation, in particular comparison to temperature which is more comparable being expressed in the same unit of measure (°C) (data not shown).

## 4. Discussion

Meteorological models’ predictions are affected by uncertainty which can be linked not only to an imperfect representation of the initial conditions of the atmosphere (small errors in the initial conditions of a forecast grow rapidly and affect predictability), but also to the approximate simulation of atmospheric processes of the state of-the-art numerical models [49,50]. Initial conditions are known approximately, and consequently two initial states only slightly differing would distinguish one from the other very rapidly as time progresses [51]. Environmental surface characteristics, such as the topography (altitude, coastline, etc.) or other soil specific characteristics (land-use, water content, soil type, etc.), are also approximated according to the horizontal resolution of the model [52]. However, it should be borne in mind that, even in very high-resolution models, the atmosphere and surface characteristics will never be as accurate as in reality (also taking into consideration that some information, e.g., land use and many other soil characteristics, is often grossly not updated and in any case mediated on horizontal resolution).

In this research, the potential of a deterministic approach in a HHWS for short range prediction was investigated. Although the verification was carried out only on 28 locations and for a limited period, BOL and MOL showed promising results in predicting the WBGT, the index selected as the heat strain indicator for the WORKLIMATE high-resolution heat–health warning system. However, the forecast skill generally progressively decreased increasing the RL. Bolam and Moloch forecasts, even if characterized by a different vertical and horizontal resolution, were overall comparable for much of the Italian explored territory, while major limits have emerged in areas with complex topography. It is well known that the representation of the territory topographic features represents one of the main problems of meteorological models. Mesinger and Veljovic [53] defined topography as “the perennial vertical coordinate problem”. While in vast plains it is rather simple to reconstruct the territory characteristics, in a more complex context (for example, mountainous areas with narrow valleys and high reliefs, areas with land-sea interface, etc.) this is much more complicated. The resolution of most meteorological models is not fine enough to represent in the required detail surface features, such as hills or mountains, and the disturbances they introduce into the airflow [54]. Our study confirmed what emerged in other model validation studies [52,55], highlighting how the best performances are generally obtained for the higher resolution with an error reduction, especially in complex topography areas. In particular, the WBGT for most of the analyzed locations was well forecasted for RL1, with an average areal value of POD1 and POD1x also far above 90%. The skill decreased for RL2 (POD2 and POD2x) to between 60% and 90%. However, the risk index was generally significantly underestimated in the bottom of the valley or near reliefs (for example Bolzano and Foligno), while it is expected to be overestimated on the highest reliefs. This problem was greater for BOLAM than for MOL_G and MOL_E, confirming the positive effect of a resolution increase. However, even assuming a further increase in resolution, it would not be possible to predict the occurrence of very local microclimates (e.g., a green lawn or an asphalted square), which people most certainly encounter in their workplace [56,57,58,59]. Some underestimation problems have also been highlighted in two coastal locations (Capo Bellavista and Palermo) where the nearest grid point model is likely located on the land-sea interface, and this problem is also well known. Lazinger [60] suggests that the issue could be solved, for example, through a linear interpolation of near grid points or using a nearest land grid point values to avoid large error.

As regards the WBGTsun, there was a general increase in the skill because the solar radiation included in the WBGT-sun calculation is much less sensitive than the other parameters to the difference in altitude between the local model and weather station.

Although forecast errors were evaluated by means of skill score, such as mean and root-mean-square error, the identification of their sources in complex models remains one of the dominating challenges [61]. With the aim to reduce the error, a comparison of the daily WBGT forecasts against the corresponding observed values, a downscaling, and a bias correction procedure were carried out by Casanueva et al. [13] in Heat Shield HHWS for 1798 locations. It is extremely difficult to hypothesize post processing correction in WORKLIMATE that have to provide forecasts for a much large number of grid points.

Another positive result of the work was that the deterioration of the forecast skill was overall low in the first three days. This aspect is very important in a HHWS addressed to vulnerable groups and in particular to the occupational sector where the activities and general actions aimed at reducing the impact of heat on workers must be planned in advance [62,63,64,65].

Despite the results highlighted, as the higher resolution models performed better in specific situations, the BOL model was used for this first version of the Worklimate operational. Since the goal of Worklimate is a five-day forecast, the use of MOL-E or MOL-G would have required the use of BOL for the fourth and fifth day, and consequently the test of different phases of the operational chain would be more complex. Furthermore, Bolam also allows simpler data management (calculation times, forecast availability time for the user, data flow management) compatible with the available resources.

The main limitation of this study was represented by the limited and unbalanced number of weather stations used in the validation (only about 28 Italian weather station were collected). Furthermore, the validation was carried out only for the May– September period between 1 July 2018 and 7 August 2020 (11 months). The validation was carried out considering the risk level according to the WBGT index thresholds calculated for a standard worker (height 175 cm, weight 75 kg), dressed without personal protective equipment, and carrying out intense activities in the sun or in the shade. Considering workers involved in different physical activities, who wear PPE and perform different duties, the results could be different in terms of categorical verification.

In the future, the model validation could be extended to other weather stations, summer seasons, and other types of workers, also increasing the spatial resolution and possibly improving the forecast by relevant end-user requirements.

## 5. Conclusions

Climate change is increasing the frequency of extreme heat wave events, necessitating the further implementation of adaptation strategies and specific interventions to safeguard worker health and productivity. At the international level, there are very few examples of personalized occupational heat health warning systems, and this study lays the foundations for the creation of a web forecasting platform and a mobile web app with customized high-resolution heat-stress-risk forecasts on the basis of worker’s characteristics, work effort, and work environment. These products are developed as part of the Worklimate project and are based on a heat stress indicator (WBGT) widely used internationally for the assessment of severe hot environments. This work assessed the performance of selected limited area models with a spatial resolution varying from 7 to 2.5 km. The results showed relatively good skills for forecasts up to three days for much of the analyzed meteorological weather locations on the Italian territory. The verification revealed promising results for the use of these models in specific warning systems for the occupational sector capable of providing information on the level of intra-daily risk. For this reason, a first experimental prototype of the system is already available on https://www.worklimate.it/en/maps-choice/shade-intense-physical-activity/ (accessed on 21 September 2021). Despite the results highlighted, with better performances of the high-resolution model, the BOL model was used for this first experimental version of the Worklimate operational system. This choice represents a good compromise between good forecast information (risk level for five daily time bands and a spatial resolution of 7 km) and a relatively easier operational chain (linked to the management of the data flow). The high temporal resolution of the selected model permits to obtain expected risk conditions on an intra-daily basis useful to better support the planning of work activity during the day based on the heat stress forecast.

In the future, further improvements in meteorological modeling, including the increase of the spatial resolution, could significantly improve the forecast, especially in complex topography areas.

Based on the results of this study, the WORKLIMATE HHWS can support the management of the occupational heat stress. It must be only considered as a system to support decisions in collaboration with existing tools that cannot in any case be separated from the direct observation of the environmental conditions of the workplace and from the individual vulnerability factors.

## Figures and Tables

**Figure 1 ijerph-18-09940-f001:**
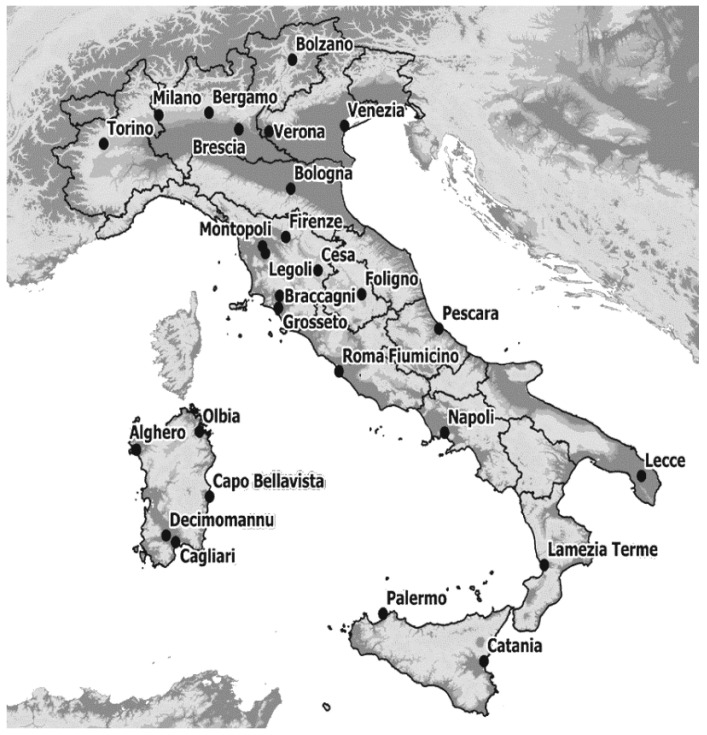
Distribution of the Italian weather stations analysed for the creation of the observation dataset. The chosen stations were identified by the name of the location.

**Figure 2 ijerph-18-09940-f002:**
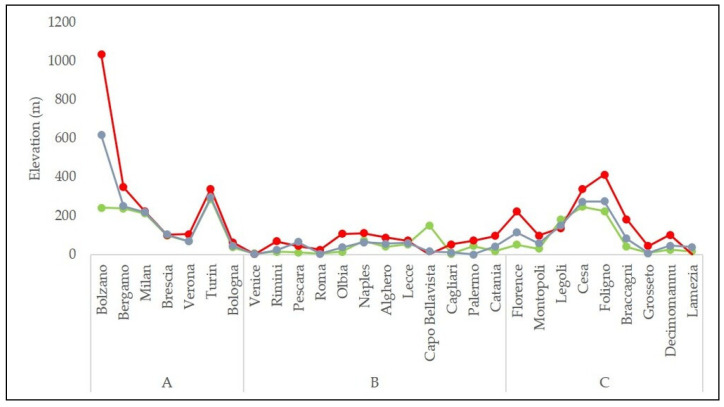
Elevation of the weather stations and the closest meteorological model grid points. Green line, weather station; Red line, BOL model; blue line, MOL model.

**Figure 3 ijerph-18-09940-f003:**
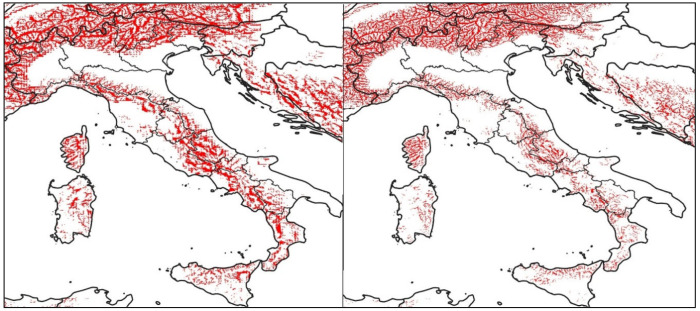
In red the areas of Italian peninsula where the models grid points (BOL on the left, MOL on the right) have an elevation higher than at least 200 m respect to that of a digital terrain model (DTM) with a spatial resolution of 90 m.

**Figure 4 ijerph-18-09940-f004:**
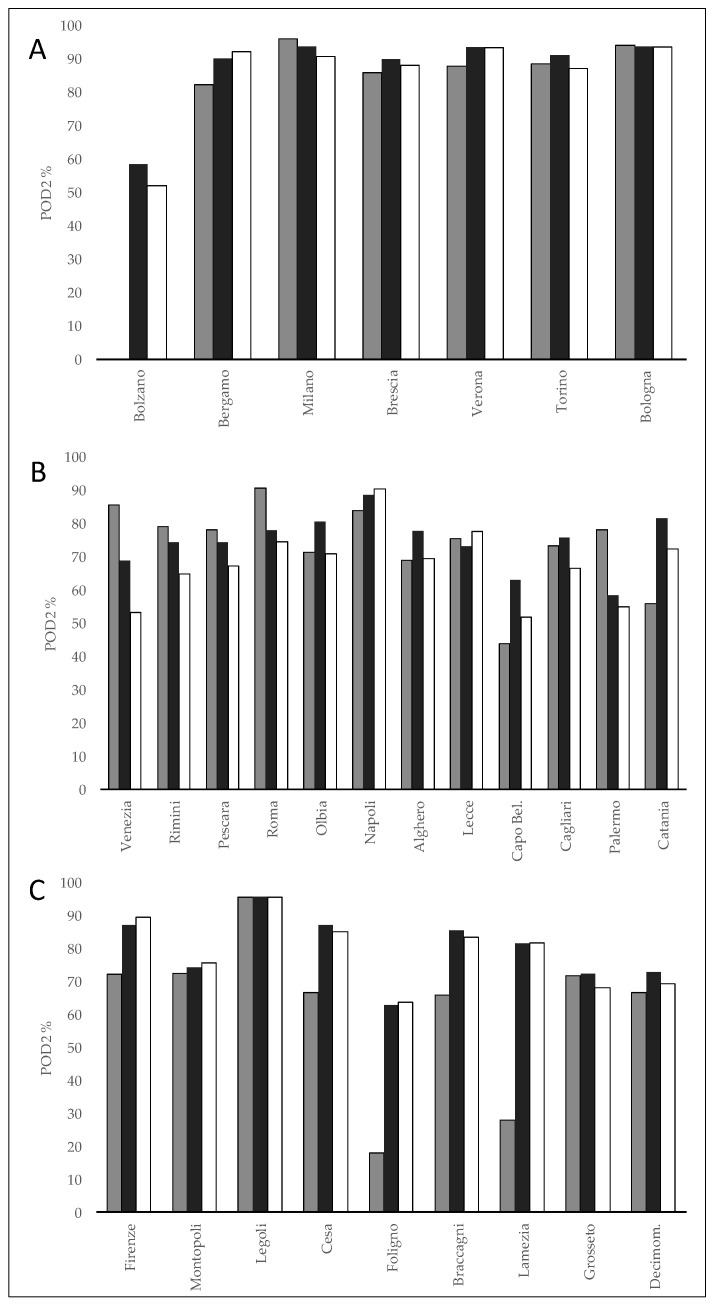
Probability of detection (POD2) of the Day2-WBGT-shade for the 12–18 time slot for each location of the three macro-geographical areas. (**A**), Northern inland plain areas; (**B**) Coastal areas; (**C**), Other central-southern inland areas; white, MOL-G; black, MOL-E; gray, BOL.

**Figure 5 ijerph-18-09940-f005:**
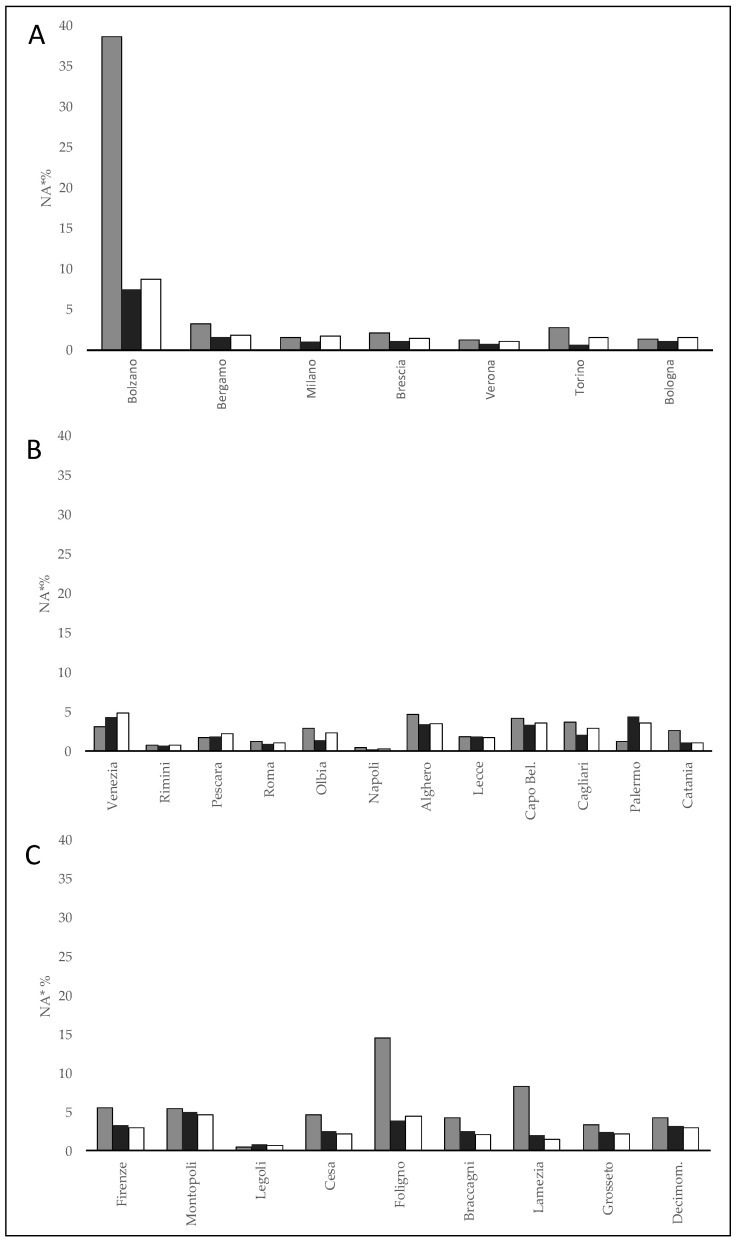
Normalized lack alarm (NA*) of the WBGT-shade for the 12–18 time band for each locality of the three macro-geographical areas. (**A**), Northern inland plain areas; (**B**), Coastal areas; (**C**), other central-southern inland areas; white, MOL-G; black, MOL-E; gray, BOL.

**Table 1 ijerph-18-09940-t001:** Distribution of the Italian weather stations used in the study. For each location, latitude (Lat.), longitude (Lon.) and altitude (Alt. a.s.l) are showed. A, North inland plain areas; B, Coastal areas; C, Central-south inland areas.

A				B				C			
Location	Lat	Lon	Alt	Localion	Lat	Lon	Alt	Localion	Lat	Lon	Alt
Bolzano	46.46	11.32	262	Venice	45.47	12.34	5	Florence	43.80	11.2	50
Bergamo	45.66	9.7	237	Rimini	44.02	12.61	13	Montopoli	43.66	10.74	29
Milan	45.63	8.72	212	Pescara	42.43	14.18	11	Legoli	43.56	10.8	180
Brescia	45.42	10.28	97	Roma	41.80	12.23	5	Cesa	43.30	11.82	246
Verona	45.38	10.87	68	Olbia	40.89	9.51	13	Foligno	42.95	12.67	224
Turin	45.20	7.64	287	Naples	40.88	14.29	72	Braccagni	42.93	11.08	40
Bologna	44.53	11.29	37	Alghero	40.63	8.28	40	Grosseto	42.74	11.05	7
				Lecce	40.23	18.13	53	Decimomannu	39.34	8.86	24
				Capo Bellavista	39.93	9.71	150	Lamezia	38.90	16.24	16
				Cagliari	39.25	9.05	3				
				Palermo	38.18	13.09	44				
				Catania	37.46	15.06	17				

**Table 2 ijerph-18-09940-t002:** Contingency table between the observed (RLO) versus predicted (RLP) values risk classes.

		RLO			
RLP		0	1	2	3
	0	C00	C10	C20	C30
	1	C01	C11	C21	C31
	2	C02	C12	C22	C32
	3	C03	C13	C23	C33

**Table 3 ijerph-18-09940-t003:** Day2-WBGT-shade average categorical skill scores for the 12–18 time slot for each geographical macro-areas. In the “northern inland plain areas”, Bolzano values were not included in the average.

	A			B			C		
Model	BOL	MOL_E	MOL_G	BOL	MOL_E	MOL_G	BOL	MOL_E	MOL_G
Data	1908	1968	1920	1902	1962	1914	1896	1956	1908
HR	82.9	79.6	80.2	80.0	79.1	78.3	74.5	78.9	79.7
CSI	78.3	75.0	75.4	75.7	74.9	73.6	69.2	75.2	75.9
POD1	81.2	78.0	79.1	84.2	82.2	84.0	79.9	80.2	81.9
POD2	89.2	92.1	90.9	73.6	74.6	67.8	61.9	79.9	79.1
POD3									
POD1x	96.2	98.1	97.2	94.9	95.4	95.0	87.8	93.8	94.4
POD2x	89.5	92.4	91.6	73.6	74.6	67.8	61.9	80.1	79.1
NA	8.5	5.3	7.0	12.0	11.0	11.5	24.3	16.9	15.0
FA	19.7	28.2	26.1	16.8	19.8	19.1	11.1	19.9	18.9
NA*	2.0	1.0	1.5	2.4	2.1	2.3	5.6	2.8	2.6
FA*	5.0	7.2	7.0	3.7	4.2	4.2	2.2	3.9	3.8
RLO 1	53.1	53.0	52.7	47.3	47.4	47.2	48.2	47.7	48.1
RLO 2	20.8	21.5	20.6	30.8	31.3	30.7	32.2	33.1	32.0
RLO 3	0.0	0.0	0.0	0.3	0.2	0.3	0.3	0.3	0.3
RLP 1	50.2	50.1	50.2	52.1	51.2	53.7	54.0	49.3	50.4
RLP 2	26.6	30.4	28.4	27.6	29.8	26.3	23.4	32.9	31.2
RLP 3	0.1	0.1	0.2	0.0	0.0	0.0	0.0	0.1	0.0

Model: BOL, BOLAM initialized on the GFS; MOL-E, MOLOCH initialized on the ECMWF; MOL-G, MOLOCH initialized on the GFS; Data, sample size; HR, hit Rate (%); CSI, critical success index (%); POD1, probability of risk level 1 detection (%); POD2, probability of risk level class 2 detection (%); POD3, probability of risk level 3 detection (%); POD1x, probability of risk level 1 or higher class detection (%); POD2x, probability of risk level 2 or higher class detection (%); NA, lack alarm (%); FA, false alarm (%); NA*, normalized lack alarm (%); FA*, normalized false alarm (%); RLO1, risk level 1 observed (%); RLO2, risk level 2 observed (%); RLO3, risk level 3 observed (%); RLP1, risk level1 predicted (%); RLP2, risk level2 predicted (%); RLP3, risk level 3 predicted (%); empty cell, it was not possible to calculate the indicator due to the lack of data observed or predicted by the model for at least one location.

**Table 4 ijerph-18-09940-t004:** Average values of Mean error, mean absolute error and root mean square error of the Day2-WBGT-shade predicted for the 12:00–18:00 time slot for the three geographical macro-areas. The scores were calculated both considering all its hourly data and the its maximum value. In the “northern inland plain areas” (A), Bolzano values were not included in the average.

	A			B			B		
Model	BOL	MOL_E	MOL_G	BOL_G	MOL_E	MOL_G	BOL_G	MOL_E	MOL_G
MAE	1.1	1.2	1.1	1.0	1.1	1.1	1.4	1.1	1.1
RMSE	1.4	1.5	1.5	1.3	1.4	1.4	1.7	1.5	1.4
ME	0.4	0.7	0.7	−0.2	−0.1	−0.1	−0.8	0.0	−0.1
Data	1908	1968	1920	1902	1962	1914	1897	1957	1909
MAEmax	1.0	1.1	1.1	1.0	1.1	1.1	1.4	1.1	1.0
RMSEmax	1.3	1.4	1.4	1.3	1.4	1.4	1.7	1.4	1.3
MEmax	0.3	0.8	0.8	−0.3	−0.1	−0.2	−0.9	0.0	0.0
Datamax	318	328	320	318	328	320	316	326	318

Model: BOL, BOLAM initialized on the GFS; MOL-E, MOLOCH initialized on the ECMWF; MOL-G, MOLOCH initialized on the GFS; MAE, mean absolute error; RMSE, root mean square error; ME, mean error; Data, sample size; MAEmax, mean absolute error of the maximum time slot value; RMSEmax, root mean square error of the maximum time slot value; MEmax, mean error of the maximum time slot value; Datamax, maximum value sample size.

**Table 5 ijerph-18-09940-t005:** Day2-WBGT-sun average categorical skill scores for the 12–18 time slot for each geographical macro-areas. In the “northern inland plain areas” (A), Bolzano values were not included in the average.

	A			B			C		
Model	BOL	MOL_E	MOL_G	BOL	MOL_E	MOL_G	BOL	MOL_E	MOL_G
Data	1902	1962	1914	1898	1958	1910	1892	1952	1904
HR	77.5	75.8	76.1	80.7	79.6	80.2	75.0	79.7	79.7
CSI	74.3	72.6	72.8	78.5	77.3	77.9	71.8	77.5	77.5
POD1	71.8	67.2	68.5	76.7	74.7	78.2	74.3	75.3	76.2
POD2	87.6	89.3	89.3	88.0	86.3	85.4	78.4	88.3	88.1
POD3									
POD1x	95.0	96.4	96.2	94.5	94.4	94.8	86.9	92.5	93.3
POD2x	91.2	94.1	93.1	88.5	87.6	86.3	78.9	89.9	89.6
NA	12.3	7.7	9.7	15.7	14.5	13.7	26.9	17.9	17.5
FA	31.4	33.4	34.3	24.6	25.4	26.2	20.1	26.4	27.0
NA*	1.8	1.0	1.3	1.8	1.7	1.6	4.3	2.0	1.9
FA*	5.7	5.9	6.4	3.5	3.4	3.7	2.9	3.7	3.8
RLO 1	41.4	41.2	41.4	35.5	34.9	35.5	36.5	35.8	36.8
RLO 2	39.8	40.2	39.4	49.2	50.2	49.0	47.9	48.7	47.4
RLO 3	0.6	0.9	0.6	1.7	1.7	1.7	2.2	2.4	2.2
RLP 1	38.7	35.8	37.3	36.4	35.6	37.9	40.3	35.3	36.8
RLP 2	45.2	48.9	47.3	51.3	52.0	49.6	44.4	51.8	50.4
RLP 3	1.8	2.5	1.9	0.4	1.0	0.7	0.5	1.5	1.1

Model: BOL, BOLAM initialized on the GFS; MOL-E, MOLOCH initialized on the ECMWF; MOL-G, MOLOCH initialized on the GFS; Data, sample size; HR, hit Rate (%); CSI, critical success index (%); POD1, probability of risk level 1 detection (%); POD2, probability of risk level class 2 detection (%); POD3, probability of risk level 3 detection (%); POD1x, probability of risk level 1 or higher class detection (%); POD2x, probability of risk level 2 or higher class detection (%); NA, lack alarm (%); FA, false alarm (%); NA*, normalized lack alarm (%); FA*, normalized false alarm (%); RLO1, risk level 1 observed (%); RLO2, risk level 2 observed (%); RLO3, risk level 3 observed (%); RLP1, risk level1 predicted (%); RLP2, risk level2 predicted (%); RLP3, risk level 3 predicted (%); empty cell, it was not possible to calculate the indicator due to the lack of data observed or predicted by the model for at least one location.

**Table 6 ijerph-18-09940-t006:** Average values of Mean error, mean absolute error and root mean square error of the Day2-WBGT-sun predicted for the 12–18 time slot for the three geographical macro-areas. The scores were calculated both considering all its hourly data and the its maximum value. In the “northern inland plain areas” (A), Bolzano values were not included in the average.

	A			B			C		
Model	BOL	MOL_E	MOL_G	BOL	MOL_E	MOL_G	BOL_G	MOL_E	MOL_G
MAE	1.3	1.4	1.4	1.2	1.2	1.2	1.4	1.4	1.4
RMSE	1.8	1.9	1.8	1.6	1.6	1.6	1.8	1.8	1.7
ME	0.7	1.0	0.9	0.0	0.1	0.0	0.1	0.5	0.5
Data	1902	1962	1914	1898	1958	1910	1905	1965	1917
MAEmax	1.1	1.2	1.2	1.1	1.2	1.2	1.4	1.3	1.3
RMSEmax	1.5	1.6	1.6	1.4	1.5	1.5	1.7	1.6	1.6
MEmax	0.4	0.9	0.9	−0.2	0.0	−0.1	0.0	0.6	0.5
Datamax	318	328	320	318	328	320	318	328	320

Model: BOL, BOLAM initialized on the GFS; MOL-E, MOLOCH initialized on the ECMWF; MOL-G, MOLOCH initialized on the GFS; MAE, mean absolute error; RMSE, root mean square error; ME, mean error; Data, Sample size; MAEmax, mean absolute error of the maximum time slot value; RMSEmax, root mean square error of the maximum time slot value; MEmax, mean error of the maximum time slot value; Datamax, maximum value simple size.

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
