# Peer review of "Performances of Limited Area Models for the WORKLIMATE Heat–Health Warning System to Protect Worker’s Health and Productivity in Italy"

_ijerph, 2021, doi:10.3390/ijerph18189940_

Round 1

Reviewer 1 Report

REVIEW  - "Performances of limited area models for the WORKLIMATE heat–health warning system: a climate change adaptation strategy to protect worker’s health and productivity in Italy"

In my opinion, the article untitled "Performances of limited area models for the WORKLIMATE heat–health warning system: a climate change adaptation strategy to protect worker’s health and productivity in Italy" is showing a very interesting initiative and so careful studied theme by a very good staff. The method sector is well explained. Some few observations:

Lines 74 -75: "The empirical WBGT according to UNI 74 EN ISO 7243 and the rational PHS according to UNI EN ISO 7933 ..." - please be careful with the acronyms use. Sometimes it is very difficult to understand the meaning of the sentences.

Lines 124-125: "To be able to meet these requirements the goal is to use, in the WORKLIMATE oper-124 ational chain, a limited area meteorological model to achieve a high spatial ..." - I think that "high spatial" could be changed to "high scale of analysis". 

Lines 127-128: "In particular, the performances of two limited area models operational at the LaMMa Consortium ..." - Is LaMMa a name or an acronym? If it is an acronym, please insert its full name there.

Table 1: I think that there is a problem between latitude and longitude data. Check it, please.

Figure 2 and 3: These figures should be inserted in color to avoid misunderstandings to the readers.

Thank you. That is a very good article in my opinion.

Author Response

For the first think, we are honored that the reviewer considers interesting our work and we understand the requests. We thanks’ to the reviewer for these observations which were appreciated and shared.

1 Lines 74 -75: "The empirical WBGT according to UNI 74 EN ISO 7243 and the rational PHS according to UNI EN ISO 7933 ..." - please be careful with the acronyms use. Sometimes it is very difficult to understand the meaning of the sentences.

Answer 1: We find your comment understandable and for this reason WGBT and PHS are now spelled out in the first occurrence in the text: “The empirical Wet Bulb Globe Temperature (WBGT) according to UNI EN ISO 7243 [16] and the rational Predicted Heat Strain (PHS) according to UNI EN ISO 7933 [17] are currently the only systems developed at an international level for an objective assessment of heat stress referring to groups of workers.”

2 Lines 124-125: "To be able to meet these requirements the goal is to use, in the WORKLIMATE operational chain, a limited area meteorological model to achieve a high spatial ..." - I think that "high spatial" could be changed to "high scale of analysis".

Answer 2: We totally agree with the reviewer and we have change "high spatial" to "high scale of analysis".

3. Lines 127-128: "In particular, the performances of two limited area models operational at the LaMMa Consortium ..." - Is LaMMa a name or an acronym? If it is an acronym, please insert its full name there.

Answer 3: LaMMa is an acronym and the reviewer's comment is understandable. The full name has now spelled out in the first occurrence in the text: “In particular, the performances of two limited area models operational at the “Environmental Modelling and Monitoring Laboratory for Sustainable Development (LaMMA Consortium)” were tested on several locations along Italy over a period of about three summer seasons and their possible use in the operational chain will be discussed highlighting strengths and weaknesses.”

4. Table 1: I think that there is a problem between latitude and longitude data. Check it, please.

Answer 4: Sorry for the mistake. we had reversed the columns "Lat" and "Long" - The error has now been fixed.

5. Figure 2 and 3: These figures should be inserted in color to avoid misunderstandings to the readers.

Answer 5: We have accepted the reviewer's request and we have updated Figures 2 and 3.

Reviewer 2 Report

In this work, the authors present the results of verification of two limited area models for possible use in thermal stress assessment in heat–health warning system. This issue is important in the context of climate change, increasing of thermal stress frequency, and the need of adaptation strategy developing for occupational health protection. The study fills the gap in spatial high-resolution initial data in the operational chain of heat–health warning system. It is fully suits the scope of International Journal of Environmental Research and Public Health.

I can suggest minor revision focusing on some points.

The authors should consider the possibility of changing the title because the adaptation strategy is not directly addressed. The verification of models is only presented.

L. 74-75 – WGBT and PHS should be spelled out because the first occurrence in the text.

L. 79 – Wet-Bulb Globe Temperature (WBGT) should be spelled out earlier.

L. 127 – Please provide references giving more information about LaMMa Consortium and its models.

Table 1 – Pease provide the unit for altitude

L. 176 – Why are there different values of Bolzano altitude (241 in the table 1 and 262 m a.s.l. in the text)?

L. 178 - This paragraph should be supplemented by references on other examples of using Bolam and Moloch models in environmental studies.

L. 179 – Please add word ‘models’ after ‘Bolam and Moloch’.

L. 178-213 – The input data should be clarified and the parametrization should be described more clearly.

L. 217-219 – How it was considered in models?

L. 259-263 – Please specify the calculation of WBGTeff, WBGT RAL and WBGT REL.

L. 261 – What parameters were used to define the acclimatized and unacclimatized workers?

Table 2. – What mean C00, C01, … C33?

L. 280 – Please provide the reference for Hit Rate as well as for other skill scores.

Fig. 4 and 5. – Why were the graphs chosen to present information on scattered locations? This can be confusing in the interpretation of the results. There are not connected to each other locations, perhaps the presentation using scatter plot will be more appropriate.

Author Response

We thank you for your appreciation on our research and we thank you even more for the suggestions because they allow us to improve the work.

1.The authors should consider the possibility of changing the title because the adaptation strategy is not directly addressed. The verification of models is only presented.

Answer 1: We welcome the reviewer's suggestion. The title is now: “Performances of limited area models for the WORKLIMATE heat–health warning system to protect worker’s health and productivity in Italy”

2. Line. 74-75: WGBT and PHS should be spelled out because the first occurrence in the text.

Answer 2: We find your comment understandable and for this reason WGBT and PHS are now spelled out in the first occurrence in the text: “The empirical Wet Bulb Globe Temperature (WBGT) according to UNI EN ISO 7243 [16] and the rational Predicted Heat Strain (PHS) according to UNI EN ISO 7933 [17] are currently the only systems developed at an international level for an objective assessment of heat stress referring to groups of workers.”

3. Line. 79: Wet-Bulb Globe Temperature (WBGT) should be spelled out earlier.

Answer 3: We find your comment understandable and for this reason WGBT is now spelled out in the first occurrence in the text.

4. Line. 127: Please provide references giving more information about LaMMa Consortium and its models.

Answer 4: We apologize for not having specified the activities carried out by the LaMMa Consortium. Information has been added in the text (line 184- sub-chapter 2.3), as well as the link to the LaMMa Consortium website: “LaMMa ((https://www.lamma.rete.toscana.it) is a public consortium between the Tuscany Region and the National Research Council which carries out activities related to observation systems and meteorological modeling at different spatial scales. Furthermore, the LaMMA provides meteorological forecasts to the Civil Protection and carries out research activities in various fields, including the climatological one.”

5. Table 1: Pease provide the unit for altitude

Answer 5: We have added the Unit for altitude.

6. Line. 176: Why are there different values of Bolzano altitude (241 in the table 1 and 262 m a.s.l. in the text)?

Answer 6: Sorry for the mistake. We replaced 241 m a.s.l. with 262 m a.s.l. in the table 1.

7. Line. 178: This paragraph should be supplemented by references on other examples of using Bolam and Moloch models in environmental studies.

Answer 7: We have added a recent paper that shows a complete review of model applications in various fields, both operational and scientific research: “Davolio et al report that Bolam end Moloch have been used for numerous scientific studies and applications, as for instance: sensitivity and impact studies, and diagnos-tics of meteorological phenomena including severe weather and storms. In addition, they report that these models were used in several operational apllications.”

8. Line. 179: Please add word ‘models’ after ‘Bolam and Moloch’.

Answer 8: We have modified the sentence according the reviewer's suggestion “Bolam and Moloch models were chosen for the comparison”.

9. Line. 178-213: The input data should be clarified and the parametrization should be described more clearly.

Answer 9: We have implemented in the text the description relating to the input data and the model parameterization with new references: “The main prognostic variables are the wind components, the absolute temperature, the surface pressure, the specific humidity and the turbulent kinetic energy. The surface layer and the planetary boundary layer are modelled accordingly to the similarity the-ory [32], with a mixing-length based turbulence closure model, to parameterize the turbulent vertical diffusion of momentum, heat and moisture. The turbulence closure is of order 1.5 [33], in which the turbulent kinetic energy is predicted. The Soil Model uses 4-6 layers, and computes surface energy, momentum, water and snow balances, heat and water vertical transfer, vegetation effects at the surface (evapo-transpiration, interception of precipitation, wilting effects etc.) and in the soil (extraction of water by roots). It takes into account the observed geographical distribution of different soil types and soil physical parameter. The atmospheric radiation is computed with a com-bined application of the global radiation [34] scheme and the ECMWF scheme [35, 36]”

10. Line. 217-219: How it was considered in models?

Answer 10: Unfortunately, the share considered by the models is not the real one of the locality. In particular, for locations with complex orography, the difference between the real altitude and that indicated by the model reaches the highest values.

11. Line. 259-263: Please specify the calculation of WBGTeff, WBGT RAL and WBGT REL.

Answer 11: The reviewer is absolutely right. The bibliographic reference number 16 is wrong, it is actually 15 (15. Morabito, M .; Messeri, A .; Noti, P .; Casanueva, A .; Crisci, A .; Kotlarski, S .; Orlandini, S. ; Schwierz, C .; Spirig, C .; Kingma, BRM; Flouris; AD; Nybo, L. An Occupational Heat-Health Warning System for Europe: The HEAT-SHIELD Platform. Int J Environ Res Public Health 2019, 16 (16 ). doi: 10.3390 / ijerph16162890.). In reference number 16, the calculations of WBGTeff, WBGT RAL and WBGT REL are explained and made explicit, however, we decided to include the formulas also in this paper, however accepting the suggestion of the reviewer.

12. Line. 261: What parameters were used to define the acclimatized and unacclimatized workers?

Answer 12: There are numerous definitions of subject acclimatized and not acclimatized to the heatAccording to previous. Some studies consider 1 to 2 weeks of daily heat exposure are needed to gain adaptation that reduces physiological strain and helps to improve physical work capabilities under a hot environment. Other studies have shown that about 75% of the physiologic adjustments occur within the first 4–6 days of heat exposure and another recent study revealed that 5 days of exposure to heat sessions were enough to acclimatize to heat workers involved in hot-climate countries.  In WORKLIMATE heat–health warning system, the 5-day threshold with critical heat stress conditions (in our case with at least a moderate risk level) was used to define when a worker can be considered acclimatized to heat within a warm season and we have added these sentence in the text.

13. Table 2: What mean C00, C01, … C33?

Answer 13: C00, C01, ..., C33 represent the possible combinations that can emerge between the risk classes predicted values (RLP) and observed values (RLO). For example, C00 is the cases number in which risk class 0 was correctly predicted. C01 is the number of times in which class 1 was predicted but instead class 0 was observed (wrong forecast).

14. Line. 280: Please provide the reference for Hit Rate as well as for other skill scores.

Answer 14: We have added the reference number "" which represents the basis of statistical analysis in the meteorological field.

15. Fig. 4 and 5: Why were the graphs chosen to present information on scattered locations? This can be confusing in the interpretation of the results. There are not connected to each other locations, perhaps the presentation using scatter plot will be more appropriate.

Answer 15: We agree with the reviewer because the line graph was not correct from a mathematical point of view. We therefore preferred to replace with histograms that highlight better the differences between the models.
